# Isolation of Strong Antioxidants from *Paeonia Officinalis* Roots and Leaves and Evaluation of Their Bioactivities

**DOI:** 10.3390/antiox8080249

**Published:** 2019-07-27

**Authors:** Lijana Dienaitė, Milda Pukalskienė, Audrius Pukalskas, Carolina V. Pereira, Ana A. Matias, Petras Rimantas Venskutonis

**Affiliations:** 1Department of Food Science and Technology, Kaunas University of Technology, Radvilėnų˛ pl. 19, LT-50254 Kaunas, Lithuania; 2IBET—Instituto de Biologia Experimental e Tecnológica, Food & Health Division Apartado 12, 2780-901 Oeiras, Portugal

**Keywords:** *P. officinalis*, phytochemicals, antioxidant capacity, cytotoxicity, α-amylase inhibition

## Abstract

*Paeonia officinalis* extracts from leaves and roots were tested for their antioxidant potential using in vitro chemical (Folin-Ciocalteu, 2,2-diphenyl-1-picrylhydrazyl radical (DPPH), 2,2′-azino-*bis*-3-ethylbenzothiazoline-6-sulfonic acid (ABTS), oxygen radical absorbance capacity (ORAC), hydroxyl radical antioxidant capacity (HORAC), hydroxyl radical scavenging capacity HOSC)) and cellular antioxidant activity (CAA) assays. Leaf extracts were stronger antioxidants than root extracts, while methanol was a more effective solvent than water in chemical assays. However, the selected water extract of leaves was a stronger antioxidant in CAA than the methanol extract (0.106 vs. 0.046 µmol quercetin equivalents/mg). Twenty compounds were identified by ultra performance liquid chromatography-quadrupole-time-of-flight (UPLC-Q-TOF) mass spectrometer, while on-line screening of their antioxidant capacity by high performance liquid chromatography (HPLC) with a DPPH^•^-scavenging detector revealed that gallic acid derivatives are the major peony antioxidants. Root water and leaf methanol extracts inhibited α-amylase in a dose dependent manner. The IC_50_ value for the strongest inhibitor, the methanol extract of leaves, was 1.67 mg/mL. In addition, the cytotoxicity assessment of extracts using human Caco-2 cells demonstrated that none of them possessed cytotoxic effects.

## 1. Introduction

Plants biosynthesize a large number of various phytochemicals that have demonstrated antioxidant and health beneficial properties in numerous studies. Polyphenolics can be defined as a group of heterogeneous biologically active non-nutrients, belonging to the most important and widely investigated class of such phytochemicals [1]. In the last few decades, numerous publications have reported that high consumption of phytochemical-rich foods might reduce the risk of several diseases. Therefore, modifying one’s diet by increasing the intake of fruits, vegetables, herbs and spices may be a promising strategy for cancer prevention.

Formation of free radicals and reactive oxygen species (ROS) is a normal process in human cells. However, excessive production of ROS, which may occur due to various exogenous factors, can disturb homeostatic conditions, resulting in oxidative stress, which may largely contribute to the development of chronic health problems, including cancer, inflammation, cardiovascular diseases and aging [2]. Therefore, it has been hypothesized that polyphenolic compounds and other dietary antioxidants, which are abundant in many fruits, vegetables, and botanicals, are essential nutrients protecting against harmful effects of the excessive free radicals [1]. Since ancient times, botanicals have been used in folk medicine to treat various diseases and health disorders, as well as in flavourings, fragrances, and colorants. However, only recently have the health benefits of many medicinal and spicy plants been explained by scientific evidence. For instance, a large number of polyphenolic antioxidants have been identified and characterized in aromatic herbs, spices, and other plant materials. Many of them have been proved as effective antioxidants [3], while their health benefits have been linked to various mechanisms, including the scavenging of harmful free radicals [1]. However, considering a vast number of species in the Plant Kingdom, there are still many poorly investigated plant species, which might be a good source of new natural bioactive substances including strong antioxidants.

Literature survey indicates that some species of the genus *Paeonia* (Paeoniaceae), which is divided into the three sections (*Moutan*, *Oneapia* and *Paeonia*) may be considered poorly studied plants [4]. The section of *Paeonia* consists of 25 species, which are widely distributed throughout temperate Eurasia. Many of them, including *P. officinalis*, remain under-investigated until now. For instance, among 262 compounds identified in different anatomical parts of *Paeonia* (terpenoids, tannins, flavonoids, stilbenes, steroids, paeonols, and phenols), only several anthocyanins were reported in *P. officinalis* flowers and one phenolic acid glycoside (1,2,3,6-*tetra*-*O*-galloyl-β-*d*-glucose) in its roots [4].

A large spectrum of bioactive substances found in different *Paeonia* spp. may be responsible for their biological and pharmacological activities. For instance, it has been used in folk medicine to treat epilepsy, liver diseases, diarrhea, and many other disorders [5]. Furthermore, antioxidant, antitumor, anti-inflammatory, anti-microbial, immune system modulation, central nervous, and cardiovascular system protective activities were also reported for *Paeonia* plants [6,7,8,9,10]. *P. lactiflora* roots are among the most traditional Chinese medicines and have been used for their anti-inflammatory, analgesic, blood tonifying, stringent, and menstruation regulation properties [11]. Besides the ornamental value of flowers, the seeds of some other *Paeonia* spp. (*P. lactiflora*, *P. suffruticosa*) have been considered as a rich source of polyunsaturated oil and proteins for foods [12]. *P. officinalis* has been used mainly for medicinal purposes, e.g., as an antiepileptic and antispasmodic drug, whereas its flowers were used to produce syrup [13].

Considering the rather scarce scientific knowledge reported for *P. officinalis*, the purpose of this study was to perform more systematic studies of its extracts, including its phytochemical composition and antioxidant activity, in order to provide more comprehensive data on the potential of *P. officinalis* as a source for functional ingredients for various applications.

## 2. Materials and Methods

### 2.1. Plant Material, Solvents and Chemicals

*Paeonia officinalis* roots and leaves of blooming plant were harvested at the Kaunas Botanical Garden of Vytautas Magnus University (Kaunas, Lithuania) and dried in a ventilated and protected from direct sunlight room. Dried leaves were separated from stems before further use.

Analytical grade methanol was purchased from StanLab (Lublin, Poland); formic acid (98%), liquid chromatography-mass spectrometry (LS-MS) and high performance liquid chromatography (HPLC) grade acetonitrile, 2,2′-azino-*bis*-3-ethylbenzothiazoline-6-sulfonic acid (ABTS, 98%), Folin-Ciocalteu reagent, monopotassium phosphate, gallic acid, 2,2-diphenyl-1-picrylhydrazyl radical (DPPH^•^, 98%), 2′,2′-azo-*bis*-(2-amidinopropane) dihydrochloride (AAPH), picolinic acid, 6-hydroxyl-2,5,7,8-*tetra*-methylchroman-2-carboxylic acid (Trolox), potassium chloride, H_2_O_2_, caffeic acid, cobalt (II) fluoride tetrahydrate, NaCl, and sodium phosphate monobasic monohydrate were from Sigma-Aldrich (Darmstadt, Germany); ethanol (99%) were from Scharlau (Barcelona, Spain); microcrystalline cellulose (20 µm) was from Sigma-Aldrich (Steinheim, Germany); sodium phosphate dibasic dihydrate, potassium iodine, and ferric chloride were from Riedel-de-Haen (Seelze, Germany); disodium fluorescein was from TCI Europe (Antwerpen, Belgium); α–amylase, type VI-B (from porcine pancreas), sodium carbonate, quercetin, and 2′,7′-dichlorofluorescin diacetate (DCFH-DA) were from Sigma-Aldrich (St. Quentin Fallavier, France); potato starch was from Fluka (Buchs, Switzerland); acarbose was from Bayer Pharma AG (Leverkusen, Germany); epithelial colorectal adenocarcinoma cells (Caco-2) were from DSMZ (Braunschweig, Germany). Trypsin, Roswell Park Memorial Institute (RPMI) 1640, penicillin streptomycin, and heat inactivated fetal bovine serum were obtained from Invitrogen (Gibco, Paisley, UK); phosphate buffer saline solution (PBS) was from Sigma-Aldrich (St. Louis, MO, USA). Ultrapure water was produced in a Simplicity 185 system (Millipore, MA, USA).

### 2.2. Extraction Procedure

Dried leaves and roots were ground in a laboratory mill Vitek (An-Der, Austria) by using a 0.5 mm size sieve. Methanol extract was prepared in an accelerated solvent extractor ASE 350 (Dionex, Sunnyvale, CA, USA) from 10 g of material, which was mixed with 4 g of diatomaceous earth and placed in a 66 mL extraction cell. Extraction was carried out three times at 60 °C temperature and 10 MPa pressure with a 15 min static and a 90 s purge time for each extraction cycle (in total, 3 cycles). Peony leaves (10 g) were also extracted in a conical flask with 200 mL methanol by using a mechanical shaker (Sklo Union LT, Teplice, Czech Republic) at room temperature and 170 rpm. Water extracts were prepared from 5 g of leaves and 2 g of roots in a conical flask by suspending plant powder in 50 mL of distilled water at 80 °C and continuously stirring at 400 rpm by magnetic hotplate stirrer (IKA, Wilmington, DE, USA). The extraction procedures of methanol and water were repeated three times, each lasting 24 h and 15 min, respectively. After methanol (TR) and water extraction the solids were filtered through a 0.3 µm filter (Filtrac, Niederschlag, Germany) and combined. Methanol was removed in a Rotavapor R-114 (Büchi, Flawil, Switzerland), then additionally dried in a flow of nitrogen (20 min) and finally all methanol and water extracts were freeze dried in a Maxi Dry Lyo (Hetto-Holton AIS, Allerod, Denmark). Solid residues after each extraction were also dried and stored at −18 °C in a freezer until further analysis. Three replicate extractions were performed for each plant, material, solvent, and method. The extracts obtained are abbreviated by the following letters: P—peony, L—leaves, R—roots, M—methanol, W—water, ASE—accelerated solvent extraction, and TR—traditional extraction.

### 2.3. Determination of Antioxidant Potential by Single Electron Transfer Based Assays

Fast colorimetric methods were selected for the in vitro assessment of total phenolic content (TPC), DPPH^•^ scavenging, and ABTS^•+^ decolourization capacity. Detailed description of these methods are provided elsewhere [14]. Briefly, for TPC, 30 µL of extract at various concentrations were mixed in a 96-well microplate with 150 µL Folin-Ciocalteu reagent diluted in distilled water (1:10 *v*/*v*) and 120 µL 7% Na_2_CO_3_ solution. After shaking 10 s the absorbance was recorded at 765 nm in a FLUOstar Omega reader (BMG Labtech, Offenburg, Germany). The calibration curve was prepared using 10–250 µg/mL solutions of gallic acid in water. The TPC was expressed in mg gallic acid equivalents (GAE) per dry weight of plant (DWP) and dry weight of extract (DWE (GAE/g DWP and GAE/g DWE, respectively) from four replicate measurements.

For DPPH^•^ scavenging 8 µL of extract and 292 µL of DPPH^•^ (6 × 10^−5^ M) solutions in methanol were mixed in a 96-well microplate and the absorbance was recorded at 515 nm in a FLUOstar Omega reader every min during 60 min. Trolox solutions (299–699 µM/L) were used for the calibration curve, the results expressed as trolox equivalents (TE) in g of DWP and DWE from 4 replicate measurements.

For the ABTS^•+^ decolourization reaction, working solution of ABTS^•+^ was produced by mixing 50 mL of ABTS and 200 µL of potassium persulfate stock solutions and kept 15 h at room temperature in the dark. The absorbance was adjusted to 0.800 ± 0.020 at 734 nm with PBS (phosphate buffered saline) and 294 µL of ABTS^•+^ were mixed with 6 µL of methanolic extract solutions in a 96-well microplate. The absorbance was measured in a FLUOstar Omega reader at 734 nm during 30 min at 1 min intervals. Calibration curve was constructed by using Trolox solutions (399–1198 µM/L), and the results were expressed as µM TE/g DWP and DWE from 6 replicate measurements.

The QUick, Easy, New, CHEap and Reproducible (QUENCHER) method was applied to determine the antioxidant capacity of solid materials before and after extractions in order to evaluate the effectiveness of the recovery of antioxidants. Solid materials were mixed with microcrystalline cellulose at a ratio from 1:5 to 1:100 (TPC and ABTS^•+^), or from 1:1 to 1:100 (DPPH^•^). For TPC, 5 mg of sample/blank (cellulose) were mixed with 150 µL of MeOH:H_2_O (1:4), 750 µL Folin-Ciocalteu’s reagent, and 600 µL Na_2_CO_3_ solution, vortexed for 15 s, shaken at 250 rpm for 3 h in the dark, centrifuged (4500 rpm, 10 min), and the absorbance of the optically clear supernatant was measured at 765 nm. For the DPPH^•^ scavenging assay, 5 mg of sample/blank were mixed with 40 µL of MeOH:H_2_O (1:4) and 1960 µL of DPPH^•^ methanolic solution, vortexed 60 s, shaken at 250 rpm for 25 min in the dark, centrifuged (for 4800 rpm, 3 min), and the absorbance of the optically clear supernatant was measured at 515 nm. For ABTS^•+^ decolorisation, 5 mg of sample/blank were mixed with 40 µL of PBS and 1960 µL of working ABTS^•+^ solution, vortexed for 60 s, shaken at 250 rpm for 30 min in the dark, centrifuged (4800 rpm, 3 min), and the absorbance of optically clear supernatant was measured at 734 nm. The values were calculated for g DWP from 6 replicate measurements.

### 2.4. Determination of Antioxidant Potential by Peroxyl-Radicals Inhibition Assays Based Hydrogen Atom Transfer

Oxygen radical absorbance capacity (ORAC), hydroxyl radical antioxidant capacity (HORAC), and hydroxyl radical scavenging capacity (HOSC) assays, which are based on peroxyl-radicals inhibition (HAT), and therefore are considered as more relevant to the processes in the biological systems [15], were selected for the characterisation of peony extracts. Detailed description of these methods are provided elsewhere [14]. Briefly, in ORAC, 25 µL of trolox standards, antioxidant, and 150 µL 2 × 10^−7^ mM of FL (fluorescein) solutions in PBS (75 mM, pH 7.4) were placed in a black 96-well microplate. Then, the mixture was preincubated in a FL800 microplate fluorescence reader (Bio-Tek Instruments, Winooski, VT, USA) for 10 min at 37 °C. To start the reaction 25 µL of AAPH (153 mM), which was used as a source of peroxyl radical, were added to each well automatically through the injector coupled with the FL800 microplate reader. The fluorescence was recorded every min during 1 h at 485 ± 20 nm excitation and 530 ± 25 nm emission. Trolox solutions (5–40 µM) were used for calibration, the results expressed in µM TE/g DWP and DWE.

For HORAC, 30 µL of sample/standard/blank, 170 µL of FL (9.28 × 10^−8^ M) and 40 µL of H_2_O_2_ (0.1990 M) solutions were pipetted into a black microplate, 60 µL of CoF_2_ (3.43 mM) solution was added, and the microplate was placed on a FL800 microplate fluorescence reader for 60 min at 37 °C using 485 nm excitation and 530 nm emission filters. The calibration curve was constructed by using 50 to 250 µM caffeic acid containing solutions. SPB (sodium phosphate buffer) (75 mM, pH 7.4) was used for hydrogen peroxide and fluorescein preparation, while acetone:Milli-Q water (50:50 *v*/*v*) was used as a blank and for the preparation of sample, calibration, and CoF_2_ solutions. Data was expressed as µM caffeic acid equivalents (CAE) per g DWP and DWE.

For the HOSC assay to 30 µL of blank/standard/sample prepared in acetone:MilliQ water (50:50 *v*/*v*), 40 µL of H_2_O_2_ (0.1990 M), and 170 µL of FL solution (9.28 × 10^−8^ M), 60 µL FeCl_3_ solution (3.43 mM) were pipetted in a black microplate, which was immediately placed in the FL800 reader, and the fluorescence was recorded every min during 1 h at 37 °C using 485 ± 20 nm excitation and 530 ± 25 nm emission filters. FeCl_3_ and H_2_O_2_ solutions were prepared in ultrapure water, while SPB (75 mM, pH = 7.4) was used to prepare the solution of FL. Trolox containing solutions (5–30 µL) were used for the calibration curve, and antioxidant capacity values were expressed as µM TE/g. Mean values in all these assays were calculated for g DWP and DWE from 4 replicate measurements.

### 2.5. Evaluation of HPLC-DPPH^•^ Scavenging On-Line

The HPLC (high performance liquid chromatography) system consisted of a Rheodyne 7125 manual injector (Rheodyne, Rohnert Park, CA, USA), a Waters 996 photodiode array detector (Milford, MA, USA), and a Waters 1525 binary pump (Milford, MA, USA). Separation of compounds was performed at 40 °C on a Hypersil C18 analytical column (5 µm, 250 × 0.46 cm, Thermo scientific, USA). The linear binary gradient was used at a constant flow rate 0.8 mL/min with solvent A (1% formic acid in ultrapure water) and B (acetonitrile) by using the following gradient elution order: 0 min 5% B, 0–2 min 10% B; 2–40 min 22% B; 40–55 min 100% B, 55–60 min 5%. 20 µL of the sample was injected and the spectra were recorded in the range from 210 to 450 nm. In order to detect radical scavengers, the HPLC system was coupled with an Agilent 1100 series pump (Agilent Technologies, USA) supplying freshly prepared methanolic DPPH^•^ (5 × 10^−6^ M) solution into a reaction coil (15 m, 0.25 mm ID) at a flow rate of 0.6 mL/min. A Shimadzu SPD-20A UV detector (Shimadzu Corporation, Kyoto, Japan) was used to record the negative peaks due to a decrease of absorbance at 515 nm after the reaction of radical scavengers with DPPH^•^. For the preliminary identification of compounds, a quadrupole mass detector Micromass ZQ (Waters, Milford, MA, USA) in negative ionization mode was used with the following parameters: cone voltage 30 V; cone gas flow 80 L/h; desolvation temperature 300 °C; desolvation gas flow 310 L/h; capillary voltage 3000 V; source temperature 120 °C; scanning range 100 to 1200 *m*/*z*. Chromatographic conditions were used as described above. Diluted extracts in methanol and water mixture (50:50, *v*/*v*) were analyzed.

### 2.6. Ultra Performance Liquid Chromatography/Electron Spray Ionisation Quadrupole Time-of-Flight Mass Spectrometry (UPLC/ESI-QTOF-MS) Analysis

UPLC/ESI-QTOF-MS analysis was carried out on a Waters Acquity UPLC system (Milford, MA) combined with a MaXis 4G Q-TOF (quadrupole time of flight) mass spectrometer, a sample manager, PDA detector, binary solvent manager, and controlled by HyStar software (Bruker Daltonic, Bremen, Germany). The Acquity C18 column (1.7 µm, 100 mm × 2.1 mm i.d. Waters, Milford, MA, USA) at a separation temperature 40 °C was used. The mobile phase consisting of solvent A (1% formic acid in ultrapure water) and solvent B (acetonitrile) was eluted in the following order: 95% A at 0.0 min; 0–5% B at 0 min; 5–20% B at 0–9 min; 20–50% B at 9–12 min, 50–100% B at 12–14 min, and held these condition for 1 min. Finally, the initial conditions were re-introduced over 1 min and held for 1 min. Before each run, the column was equilibrated for an additional 1 min. The following parameters were: a capillary voltage of 4 kV; an end plate offset of −0.5 kV; a flow rate of drying-gas of 10.0 L/min; a nebulizer pressure of 2.5 bar; a scanning range 79–2400 *m*/*z*; an injection volume of 1 µL; a flow rate of 0.45 mL/min. For fragmentation study, a data dependent scan was performed by deploying collision induced dissociation (CID) using nitrogen as a collision gas at 30 eV. Peaks were identified and analysed by comparing their retention times, accurate masses, and formulas by using external standards and commercial databases.

Quantitative analysis was performed by using external standards. Calibration curves were drawn using six concentrations of standard solutions and represented the dependence between the integrated chromatographic peak areas and the corresponding amounts of injected standards. According to the lowest point of the calibration curve, the LOQ (S/N = 10) and LOD (S/N = 3) were calculated.

### 2.7. Determination of α-Amylase Inhibitory Activity

An α-Amylase assay was carried out according to the procedure of Al-Dabbas et al. [16]. Briefly, 60 µL of blank (PBS)/sample/acarbose (0.02 mg/mL) solutions and 200 µL of a starch solution (400 µg/mL) were mixed in 6 Eppendorf tubes for incubation at 37 °C for 5 min. To start the reaction, twenty µL of α-amylase (5 mg/mL) was added to the three tubes and in the rest of three twenty µL of PBS (pH = 7.4) as a control for the sample. Afterwards, twenty µL of PBS was added to all tubes and incubated at 37 °C for 7.5 min. After incubation, in order to determine the degradation of the starch, 200 µL of I_2_ solution (0.01 M) was transferred to the Eppendorf tubes. Finally, by adding 1 mL of a distilled H_2_O reaction, the mixture was diluted and its absorbance was measured at 660 nm by using a GENESYS 10 spectrophotometer (Thermo Scientific, Waltham, MA, USA). For the calibration, a curve stock solution of acarbose (5 mg/mL) was used. The inhibition of enzyme activity (IC_50_) was expressed as mg/mL. The experiments were performed in triplicate.

### 2.8. Cytotoxicity Assay in Caco-2 Cells

The cells were maintained as monolayers in 175 cm^2^ culture flasks containing an RPMI 1640 medium supplemented with 10% fetal bovine serum and 1% penicillin-streptomycin, at 37 °C in humidified air with 5% CO_2_. Caco-2 cells were seeded at a density of 2 × 10^4^ cells/well in transparent 96-well plates and allowed to grow as a confluent and non-differentiated monolayer that can mimic the human intestinal epithelium. This cell model shares some characteristics with crypt enterocytes, and thus it has been considered to be an accepted intestinal model widely implemented to assess the effect of chemical, food compounds, and nano/microparticles on the intestinal function [17]. The medium was changed every 2 days. On the day of the experiment, cells were washed twice with pre-warmed PBS at ~37 °C. Water and ethanol extracts of peony were solubilized in H_2_O and EtOH, respectively. All extracts were prepared with a final concentration of 16.7 mg/mL. Cell-based assays were performed using a maximum concentration of solvent—50% and 5% for H_2_O and ethanol, respectively.

Cytotoxicity was assessed by the MTS (3-(4,5-dimethylthiazol-2-yl)-2,5-dyphenyltetrazolium bromide) method [17]. Prepared cells monolayers were incubated with various concentrations of peony extracts (100 µL) for 4, 24, and 48 h and washed with PBS. Their viability was determined by adding a 100 µL 10-fold diluted MTS reagent (according to the manufacturer’s guidelines), incubating for 2.5 h at 37 °C 5% CO_2_, measuring the absorbance at 490 nm in an Epoch Microplate Spectrophotometer (Bio-Tek, Instruments, Winooski, VT, USA). Data were expressed in a cellular viability percentage relative to control (%). The experiments were performed in triplicate.

### 2.9. Cellular Antioxidant Activity Assay (CAA)

The CAA assay was carried out by the procedure of Wolfe and Liu [18]. Cell monolayers were treated with 50 µL of PBS, sample and standard (quercetin, 2.5–20 µM) solution, and fifty µL of a DCFH-DA solution containing 50 µM of reagent and pre-incubated at 37 °C, 5% of CO_2_ for 1 h. Afterwards, to each well containing standards/samples and tree wells containing PBS (control) was added 100 μL of AAPH (12 mM). In the rest of the six wells containing PBS (blank) 100 μL of PBS solution was added. The data were recorded every 5 min during 60 min by using a FL800 fluorescent reader (ex. 485 nm, em. 540 nm). CAA values were expressed as µM of quercetin equivalents (QE) per g of the extract from three independent experiments.

### 2.10. Statistical Data Handling

All results are presented as means ± standard deviation (SD), and all the experiments were completed at least three times. Significant differences among means were evaluated by one-way ANOVA, using the statistical package GraphPad Prism 6. Duncans’ posthoc test was used to determine the significant difference among the treatments at *p* < 0.05.

## 3. Results

### 3.1. Total Yield, Total Phenolic Content and Radical Scavenging Capacity of Peony Extracts

The yields and antioxidant capacity values of extracts and plant materials are presented in Table 1. High polarity solvents, methanol, and water were compared in this study. In addition, methanol was applied in a pressurized liquid extraction (PLE). Methanol has been shown to be an effective solvent for polyphenolic antioxidants in numerous studies, while water is very attractive in terms of its favourable green chemistry principles and low cost. It may be observed that the total yields from leaves were remarkably higher than those from the roots in the case of both solvents, whereas PLE with methanol resulted in an approximately 10% higher yield.

Using several antioxidant activity assays is important for completing a more comprehensive evaluation of natural products. Thus, Huang et al. [15] recommended applying at least two single electron transfer (SET) and one hydrogen atom transfer (HAT) assays for this purpose. Following this recommendation, TPC (Folin-Ciocalteu), ABTS^•+^ and DPPH^•^ scavenging, as well as ORAC, HORAC, and HOSC assays, were used for measuring the antioxidant potential of the dried extracts (DWE) and recovery of antioxidants from the initial plant material (DWP). These values are useful for evaluating the antioxidant potential of extracts and the recovery of antioxidants from the raw material, respectively. For instance, a low yield extract may possess stronger antioxidant activity, while for high yields, a better recovery of antioxidants from the plant may be achieved, although the extracts may show lower antioxidant capacities due to the dilution of the active constituents with neutral ones.

Thus, TPC values measured for *P. officinalis* extracts (Table 1) were in the range of 215.7 (PRTRW)–601.1 (PLTRM) mg GAE/g DWE, while the recovery of polyphenols from the raw material was in the range of 41.42 (PRTRW)–285.9 (PLTRM) mg GAE/g DWP. Methanol extraction resulted in higher TPC values both for DWP and DWE, while the TPC in leaves was higher than that in roots. It may be observed that higher TPC values for leaves were found via conventional extraction (TR) than via PLE; most likely, this difference was due to the dilution of Folin-Ciocalteu reactive substances in the latter case (PLE gave higher yields than TR) and some other changes. Consequently, in terms of TPC, traditional methanol extraction was the most effective method.

DPPH^•^ and ABTS^•+^ scavenging assays, which are also based on SET (single electron transfer) reaction, gave similar results for the antioxidant capacity of extracts and the recovery of radical scavengers (Table 1). However, ABTS^•+^ decolourisation values were remarkably higher than DPPH^•^ scavenging values, which may be explained by the different reaction conditions [15]. Thus, DPPH^•^ and ABTS^•+^ scavenging values were in the ranges of 343.4 ± 6.06–2553 ± 28.40 and 886.0 ± 7.36–4610 ± 18.70 μM TE/g DWE, respectively, while antioxidant recoveries were 65.94 ± 1.16–1153 ± 11.25 and 170.1 ± 1.41–2151 ± 12.72 μM TE/g DWP, respectively. It should be noted that the TPC and ABTS^•+^/DPPH^•^ scavenging capacity was not reported for *P. officinalis* previously. Some antioxidants may be strongly bound to the insoluble plant matrix and are not available for any solvent without pre-treatment [19]. To evaluate the recovery of the active constituents, the antioxidant capacity of peony solids was monitored by using the so-called QUENCHER procedure [20]. The results presented in Figure 1 show antioxidant capacity values of peony solids before and after extraction. It may be observed that after extraction, they are remarkably reduced, indicating that the extraction processes were quite efficient for the recovery of antioxidants. Water was slightly more efficient than methanol.

### 3.2. Peroxyl and Hydroxyl Radicals Inhibition in ORAC, HORAC and HOSC Assays

Based on TPC and ABTS^•+^/DPPH^•^-scavenging values and chemical composition, methanol (PLASEM, PLTRM) and water (PLTRM) extracts were tested (Table 1). ORAC values varied in the ranges of 1232–1433 µmol TE/g DWE and 409.6–681.4 µmol TE/g DWP, HOSC in the ranges of 1957–2012 µmol TE/g DWE and 668.2–931.0 µmol TE/g DWP, and HORAC in the ranges of 1566–1891 µmol CAE/g DWE and 520.3–899.4 CAE/g DWP. These assays confirmed that methanol extracts were better antioxidants than water extracts. It may be observed that regardless of significant differences in antioxidant activity between some tested extracts, these differences were not remarkable. However, due to the differences in the extract yields, the recoveries of antioxidants from DWP were in a wider range. Thus, methanol extracted antioxidants more efficiently. For instance, for the ASE methanol extracted from 1 g of dried peony leaves, the quantity of antioxidants was equivalent to 0.17–0.23 g of Trolox.

### 3.3. Determination of Phytochemicals by UPLC-Q/TOF

In total, 23 compounds were detected and most of them were identified based on the measured accurate masses, suggested in various databases formulas, chromatographic retention times, MS/MS fragmentation, data obtained with authentic reference compounds, and various literature sources (Table 2, representative chromatogram in Figure 2A).

MS data and retention times of compounds **1** and **4** well matched the standards of quinic and gallic acids. The compound **2** with molecular ion *m*/*z* = 341 [M − H]^−^ and fragments of 191, 149, and 89 was assigned to dihexose. The molecular ion of compound **3** (*m*/*z* = 331.0671) corresponds to C_13_H_15_O_10_, while the fragment ion (*m*/*z* = 169.0140) fits C_7_H_5_O_5_, the residue of gallic acid. Based on the recorded *m*/*z* and literature data [21], the compound was tentatively identified as galloylhexose. The main ion of peak **5** (*m*/*z* = 321.0253) corresponds to C_14_H_9_O_9_, while the MS/MS fragmentation, due to the loss of [M − H − 152]^−^ and [M − H–CO_2_]^−^, gave the ions of *m*/*z* = 169 (gallic acid, C_7_H_5_O_5_) and *m*/*z* = 125.0240 (C_6_H_5_O_5_), respectively. This peak was assigned to digallic acid, while compound **6,** which gave *m*/*z* = 183.0302 ([M − H]^−^), was tentatively identified as methyl gallate [21]. The MS^2^ spectrum of this compound produced characteristic ions at *m*/*z* = 168.0060, 140.0112, and 124.0166. Compound **7** gave a molecular ion [M − H]^−^ (*m*/*z* = 635), and fragments of 483, 465, and 169, fitting C_27_H_23_O_18_, C_20_H_19_O_14_, C_20_H_17_O_13_, and C_7_H_5_O_5_, respectively. The loss of 152, 170, and 466 amu was attributed to the loss of [M−H–galloyl]^−^, [M − H–galloyl–H_2_O]^−^ and [M − H–2 galloyl–hexose]^−^ units, respectively. As a result, this compound was tentatively identified as *tri*-galloyl-hexoside. Compounds **8** (t_R_ 4.5), **22** (t_R_ 9.9), and **23** (t_R_ 10.2) with *m*/*z* = 491.1768, 621.0637, and 697.0695, matching C_21_H_31_O_13_, C_15_H_25_O_26_, and C_31_H_21_O_19_, respectively, have not been identified. Compounds **9** and **18** displayed a molecular ion [M − H]^−^ (*m*/*z* = 525, C_24_H_29_O_13_) and several fragment ions in the MS/MS mode. The ion at *m*/*z* 479.1508/4791358 (C_23_H_27_O_11_) was a basic fragment arising from the loss of [M − H–H_2_O–CO]^−^ (46 mass units), which, by a further loss of CH_2_O, C_7_H_6_O_2_, and [M − H–paeonyl–CHOH–2H]^−^, produced the fragments at *m*/*z* 449.1448/449.1440 (C_22_H_25_O_10_), 357.1191/357.1088 (C_16_H_21_O_9_), and 283.0818/283.0714 (C_13_H_15_O_7_), respectively. The ion at *m*/*z* 327.1086/327.1075 (C_15_H_19_O_8_) can be derived by the loss of the C_8_H_8_O_3_ from the basic ion, with *m*/*z* = 449, or from the ion with *m*/*z* = 357 (−30 mass unit; CH_2_O). Finally, the fragments corresponded to the paeonyl (*m*/*z* = 165.0556/165.0544, C_9_H_9_O_3_) and benzoyl (*m*/*z* = 121.0294/121.0281, C_7_H_5_O_2_) units resulting from the cleavage of the paeoniflorin. The compounds shared the same *m*/*z* at 121, 165, 283, 327, 357, and 449 as previously reported for paeoniflorin [22]. Considering that molecular ion (*m*/*z* = 479) of **9** and **18** was higher by 46 Da, which can be attributed to CO (28 Da) and H_2_O (18 Da), the peaks tentatively identified as carboxylated and hydratated derivatives of paeoniflorin.

Compound **10** (t_R_ 5.9 min) gave *m*/*z* = 787.1006 (C_34_H_27_O_22_), which was assigned to *tetra*-galloyl-hexoside because the main fragment at *m*/*z* = 617.0793 (C_27_H_21_O_17_) may be obtained by the loss of gallic acid (*m*/*z* = 169.0139, C_7_H_5_O_5_) residue from a deprotonated parent ion (*m*/*z* = 787.1006). The other characteristic fragment (*m*/*z* = 456.0683) may result due to the loss of the [M−H–hexosyl]^–^ moiety from *m*/*z* = 617.0793. Compound **11** was tentatively identified as quercetin-dihexoside: its molecular ion *m*/*z* = 609.1460 corresponds to C_27_H_29_O_16_), [M − H − 463]^−^ indicates the loss of quercetin-hexose (C_21_H_19_O_12_), [M − H − 146]^−^ is characteristic to hexose moiety, while *m*/*z* = 301.0325 represents quercetin (C_15_H_9_O_7_). The compound **12** with a major ion of *m*/*z* = 615.0993 was assigned to quercetin-galloyl-hexoside; the loss of 463 Da indicates quercetin hexose moiety, while the fragments of *m*/*z* = 301.0324 (C_15_H_9_O_7_) and *m*/*z* = 169.0132 (C_7_H_5_O_5_) are characteristic of quercetin and gallic acid, respectively [23]. Compound **13** was assigned to quercetin pentoside (*m*/*z* = 433.0779, [M − H]^−^), which in MS/MS fragmentation lost pentose (132 Da) and gave a fragment ion of *m*/*z* = 301.0340 matching molecular formula (C_15_H_9_O_7_) of quercetin [21]. The full scan mass spectra of compound **14** (methyl digallate) showed mainly an intense ion at *m*/*z* 335, which yielded an MS^2^ ion at *m*/*z* 183 (methyl gallate) corresponding to the loss of a gallic acid moiety ([M − H − 152]^−^). Compound **15** (t_R_ = 7.8 min) gave an [M − H]^−^ ion with *m*/*z* = 939.1122, indicating a molecular formula of C_41_H_31_O_26_. Its MS^2^ spectrum contained four fragment ions: *m*/*z* = 769.0901 (C_34_H_25_O_21_) showing the loss of [M–152 − H_2_O]^−^ and matching *tetra*-galloyl-hexose moiety; *m*/*z* = 617.0795 (C_27_H_21_O_17_), showing the loss of additional galloyl moiety [M − H − 152]^−^ and matching *tri*-galloyl-hexose moiety; *m*/*z* = 447.0569, indicating the split of [M − H − 152–H_2_O]^−^ and indicating a *di*-galloyl-hexose unit; *m*/*z* = 169.0132 (C_7_H_5_O_5_) matching gallic acid. All these data led to the identification of *penta*-galloyl-hexoside. Compound **16** (t_R_ 8.2 min) with the molecular ion of *m*/*z* = 629.1149 (C_29_H_25_O_16_) was tentatively identified as isorhamnetin-galloyl-hexoside [21]. It produced a fragment ion of *m*/*z* = 477.1046 [M − H − 152]^−^ due to the loss of gallic acid; the fragment ion of *m*/*z* = 315.0568 indicates the loss of galloyl + hexose [M − H − 152 − 162]^−^, and finally the ion of *m*/*z* = 169.0141 indicates on the presence of gallic acid. Three compounds, **17**, **19**, and **20** (t_R_ 8.4, 8.7 and 9.3 min) with the precursor ion of *m*/*z* = 545 were assigned to dihydroxybenzoic acetate-digallate derivatives based on QTOF-MS and previously reported data [21]. In addition, their spectra showed the product ions of *m*/*z* = 469.0489/469.0407/469.0381 [M − H − 76]^−^ and 393.0466/393.0461/393.0468 [M − H − 152]^−^, corresponding to the neutral losses of acetyl + H_2_O and galloyl moieties from the parent ion (*m*/*z* = 545), respectively. Finally, a fragment ion of *m*/*z* = 169.0135/169.0139/169.0140 indicates the presence of gallic acid. Compound **21** gave a [M − H]^−^ ion with *m*/*z* = 1091.1236 and similar to *penta*-galloyl-hexoside fragmentation pattern with a difference of 152 amu, suggesting an additional galloyl unit. Moreover, the presence of product ions of *m*/*z* = 939.1101, 769.0895, 617.0811, and 169.0143 suggests the presence of 6 galloyl groups. Consequently, compound **21** was tentatively identified as *hexa*-galloyl-hexoside.

### 3.4. Determination of Antioxidants by the On-Line HPLC-UV-DPPH^•^-Scavenging

HPLC-UV with the on-line DPPH^•^-scavenging detectors was used for the preliminary screening of antioxidant phytochemicals in paeony extracts. As it may be judged from the size of the negative peaks in the chromatogram (Figure 2B), gallic acid derivatives (**3**, **4**, **5**, **6**, **7**, **10**, **12**, **14**, **15**, **16**, **17**, **19**, **20**, **21**), quercetin derivatives (**11**, **13**), paeoniflorin derivatives (**9**, **18**), and unknown compounds (**22**, **23**) were the strongest radical scavengers in the investigated extracts. Compound **5** was detected only in the PLTRW extract (data not shown).

The constituents, for which reference compounds were available, were quantified and their amounts were expressed in mg/g DWE and mg/g DWP (Table 3). The concentration of quinic acid, gallic acid, and quercetin dihexoside in different peony fractions was in the ranges of 2.14–7.58, 0.44–12.33, and 1.04–1.64 mg/g DWE, respectively. Quinic acid was quantitatively a major constituent in *P. officinalis* extracts (except for PLTRW), while the amounts of quercetin *di*-hexoside and gallic acid varied depending on extract type. For instance, the PLTRW extract contained the highest amount of gallic acid (12.33 ± 0.87 mg/g DWE). Its recovery was 4.10 ± 0.25 mg/g DWP, while the best recovery of quercetin *di*-hexoside was found in case of PLTRM.

### 3.5. α-Amylase Inhibitory Properties of Selected Extracts

*P. officinalis* extracts applied in a concentration range of 1.25–5 mg/mL inhibited α-amylase in a dose dependent manner, whereas the differences for lower concentrations (0.83–1.25 mg/mL) in most cases were not significant (Figure 3). The PLTRM extract was a slightly, although significantly, stronger inhibitor of α-amylase than other leaf extracts in the concentration range of 0.83–2.5 mg/mL. The values calculated in mg/mL IC_50_ (inhibitory concentration) were in the following range: PLTRW (2.52 ± 0.32) > PLASEM (2.34 ± 0.18) > PLTRM (1.67 ± 0.17) > acarbose 0.3 ± 0.12 ^d^. Consequently, the strongest α-amylase inhibitor PLTRM extract was 5.5 times less effective than acarbose.

### 3.6. In Vitro Cytotoxic and Cellular Antioxidant Activity (CAA) Activity of Extracts

The cytotoxic effect was assessed on the Caco-2 cell line using the MTS method, where results showed no cytotoxic effect for each peony extract (PLTRW, PLASEM, PLTRM) in the range of the applied concentrations at 4 h, 24 h, and 48 h treatment (data not shown). Therefore, all extracts could be assessed in further research work. CAA-values for the peony extracts ranged from 0.046 to 0.106 μmol QE/mg extract (Figure 4).

## 4. Discussion

The total extraction yield, the concentration of target compounds in the extracts, and their recovery from the plant materials are important process characteristics. Very high yields obtained with methanol (>43%) indicate the considerable content of soluble high polarity substances in *P. officinalis* leaves (Tale 1). The yields from plant roots were almost two-fold lower than those from leaves. For instance, Bae et al. [24] reported high yields of methanol extracts for the roots of other *Paeonia* species, namely *P. lactiflora* (30%) and *P. obovate* (32.5%), obtained during sonication, while in our study, the *P. officinalis* methanol extract obtained by PLE yielded 23.89%. Other studies [25,26] reported approximately two-fold lower yields from *P. officinalis* roots obtained at room temperature with ethanol and water than in our study, most likely, due to the differences in extraction temperature, which in our study for water extraction was 80 °C. It is well known that heat is one of the most important factors for extraction rate and yield.

The assessment of antioxidant capacity in vitro is widely used for preliminary evaluation of the beneficial properties of botanical extracts [27], and, although these values are not appropriate for direct assessment of physiological bioactivities, the experts in this area recently concluded that these assays, as low-cost and high throughput tools, cannot be ignored [28]. Plants are very complex biological structures containing a large diversity of antioxidants with different chemical structures and properties. Therefore, for a preliminary screening of antioxidant activity of plant preparations, several fast and simple in vitro assays have been developed and widely used. Basically, they may be classified into single electron (SET) or hydrogen atom (HAT) transfer methods [15]. The former is based on the ability of antioxidants to scavenge radicals (e.g., ABTS^•+^ and DPPH^•^) or reduce the compounds present in the reaction mixture (e.g., FRAP, Folin-Ciocalteu) by transferring one electron to them. The HAT method is based on inhibiting peroxyl-radicals (e.g., ORAC, HORAC, HOSC), and, therefore, they are more relevant to the processes in the biological systems [15]. Due to the different reaction conditions and mechanisms, the antioxidant capacity values determined by these assays may significantly differ; therefore, there is no perfect method for the comprehensive evaluation of antioxidant potential of complex biological systems [29].

Most recently, Camargo et al. (2019) [30] also recognized colorimetric methods as important screening tools by proving the links between TPC, ORAC, and FRAP values and the anti-inflammatory potential of grape by-products, which inhibited the activation of NF-κB in RAW 264.7 cells. Kleinrichert and Alappat [31] reported that the antioxidant efficacy of plant extracts positively correlated with the disruption of A-aggregation, which is accepted as an important biomarker in the pathogenesis of Alzheimer’s Disease.

Li et al. [32] reported a remarkably lower antioxidant capacity for aqueous extracts of *P. lactiflora* and *P. suffruticosa* than the capacity determined in our study for *P. officinalis*. In the ABTS^•+^ scavenging assay, they were 365.72 ± 5.08, 85.25 ± 1.36, and 243.12 ± 4.65 μM TE/g DWE, respectively; in the TPC assay, they were 31.48 ± 1.52, 21.38 ± 0.26 and 22.37 ± 1.17 mg GAE/g DW, respectively. Cai et al. [33] reported even lower values for methanolic and aqueous root extracts of *P. lactiflora* and *P. suffruticosa*: 11.4/29.0 (methanol extracts) and 7.0/12.5 (aqueous extracts) mg GAE/g DW and 4.1/4.23 (methanol extracts) and 1.5/8.76 (water extracts μM TE/g DW, respectively. It may be assumed that these differences depend on the plant species, its anatomical parts, and the extraction procedure.

In general, in vitro SET based assays revealed the strong antioxidant potential of *P. officinalis* leaves and roots (Table 1). The highest values measured for 1 g of leaf extract were equivalent to approximately 0.64 and 1.15 g of trolox in DPPH and ABTS assays, respectively. In addition, the values obtained in the all three assays showed a good positive correlation, e.g., the extract with the highest TPC (601.1 mg GAE/g DWE) was also the strongest DPPH^•^/ABTS^•+^ scavenger. This correlation was reported in many previously published articles [34].

Dietary antioxidants are hypothesised to be possible exogenous protective agents in neutralising excessive reactive oxygen species (ROS), which may form in human organisms due to various factors. Among numerous in vitro antioxidant activity evaluation methods, peroxyl and hydroxyl radical inhibition assays (ORAC, HORAC, and HOSC) are recognised as better related to the antioxidant processes happening in the biological samples [35,36]. However, only a few studies performed ORAC, HORAC, and HOSC assays for peony extracts. For instance, Soare et al. [37] reported remarkably lower ORAC values for *P. officinalis* methanol and ethanol extracts (480.87 and 555.2 µmol TE/g, respectively). It has been shown in numerous articles that antioxidant activity may depend on various factors, such as harvesting time and climatic conditions.

The identification and characterization of individual bioactive constituents are among the most important tasks for valorising new plant materials. Bioactives in *P. officinalis* extracts were analysed by the UPLC-Q/TOF method (Table 2, Figure 2). It may be observed that gallic acid and its derivatives are the most abundant constituents in the *P. officinalis* plant (Table 3), as well as in other peony species [22,24,38]. For instance, Bae et al. [24] reported that the content of gallic acid in *P. lactiflora* and *P. obovate*, depending on plant collection site, varied from 2.4 ± 0.0 to 1740.4 ± 35.3 and from 1.6 ± 00 to 1426.2 ± 1.9 mg/100 g DW, respectively. Paeoniflorin and its derivatives were also among the main phytochemicals in peony seeds from various *Paeonia* species collected from different areas [39,40,41,42].

Type 2 diabetes mellitus is a progressive metabolic disorder of glucose metabolism, which could be treated by decreasing postprandial glucose levels. The enzyme α-amylase is responsible for the breakdown of complex polysaccharides into disaccharides. The inhibition of this enzyme could prolong overall carbohydrate digestion time, causing a reduction of glucose level in postprandial plasma. Consequently, α-amylase inhibitory activity is a useful indicator of the bioactivity of plant phytochemicals (Figure 3). To the best of our knowledge, α-amylase inhibitory activity was not previously reported for *P. officinalis* extracts.

New natural extracts or purified phytochemicals from poorly studied plants after comprehensive evaluation of their toxicity and safety may be promising disease risk reducing agents or even a platform for designing new nutraceuticals and medicines. For this purpose, cytotoxicity assays in human cells have been widely used. For instance, numerous studies have shown that many drugs used in medical treatments, such as chemotherapy, are cytotoxic to normal cells. Therefore, the search for effective compounds against cancer cells is necessary. Our results showed no cytotoxic effect for each tested peony extract. To the best of our knowledge, there are no reports on the cytotoxicity of *P. officinalis* extracts. In the literature regarding the cytotoxicity of the genus *Paeonia*, we found that Lin et al. [43] determined that *Radix Paeoniae Rubra* extract, on normal urothelial SV-HUC-1 cells, was cytotoxic at 3.5 mg/mL at 48 h and at > 3.5 mg/mL at 24 h, while the cytotoxic effect on BFTC 905 and MB49 cancer cells was 1.4 and 2.8 mg/mL and 1.4 and 1.8 mg/mL at 48 and 24 h treatment, respectively. Furthermore, Lin et al. [44] observed cytotoxicity in normal SV-HUC-1 cells for *Cortex Moutan* extract (>3.5 mg/mL at 24 h and 1.6 mg/mL at 48 h). According to Almosnid et al. [45], *cis* and *trans*-suffruticosol D isolated from *P. suffruticosa* seeds was cytotoxic against A549 (lung), BT20 (breast), MCF-7 (breast), and U2OS (osteosarcoma) cancer cell lines at 9.93 to 46.79 μM concentrations and above, while on normal breast epithelial cells (HMEC) and normal lung epithelial cells (HPL1A; EC_50_ values were in range from 146.3 to 269.5 μM and from 78.3 to 177.5 μM, respectively), the cytotoxicity wasthe notably weaker. As can be seen, a cytotoxic effect on normal cells compared to cancer cells appears after applying considerably higher concentrations of various extracts derived from plants, demonstrating that natural phytochemicals derived from medical herbs are a promising source for the development of cancer drugs.

One of the disadvantages of in vitro colorimetric antioxidant assays is that they do not provide any information about the absorption and metabolism of antioxidants in a cellular environment. Therefore, the values measured by chemical assays often do not correlate with the antioxidant processes both in vitro and in vivo, meaning that strong radical scavenging capacity values are not always confirmed by biological activities [46]. To improve the biological relevance of antioxidant activity results, the cellular antioxidant activity (CAA) method was developed for evaluating phytochemicals that can penetrate the cell membrane and prevent oxidation [18]. This method gives additional information about the uptake, absorption, and metabolism in cell environments. For instance, the strong antioxidant activity by plant polyphenolics also demonstrated numerous protective effects against chronic diseases [47]. After a preliminary evaluation of cytotoxicity, the CAA method was used for a more comprehensive evaluation of the antioxidant potential of *P. officinalis* extracts (Figure 4). This technique is more biologically relevant than chemical antioxidant activity assays, because it accounts for the uptake, metabolism, and localization of antioxidant compounds within cells [18]. It is interesting to note that the water extract demonstrated higher CAA than the methanol extracts isolated both by conventional and PLE methods (PLTRM and PLASEM), while in SET and HAT based antioxidant capacity assays, methanol was a more effective solvent than water. Similarly, water extracts were remarkably stronger antioxidants than methanol extracts in the CAA assay in our previous study with *Nepeta* spp. [14]. This result is possibly linked with the higher affinity of water extracts to aqueous cellular environments, which might be favourable for crossing the cell membrane via water-soluble phytochemicals. Xiong et al. [48] and Huang et al. [49] reported CAA values of 26.7 ± 3.0 and 18.11 ± 1.84 µmol QE/100 g of flower, respectively, for *P. suffruticosa*, which was extracted with acetone.

## 5. Conclusions

In general, this study demonstrated that *P. officinalis* is a rich source of polyphenolic compounds with high antioxidant potential. Leaf extracts were stronger agents than root extracts, while methanol was a more effective solvent than water, as was demonstrated via its chemical antioxidant capacity and in α-amylase inhibitory assays. However, the selected water extract of leaves was a stronger antioxidant in the cellular antioxidant activity assay than the methanol extracts. These results confirm that plant extracts do not necessarily have the same response in a biological environment (in vitro) as chemical assays. Additionally, none of the extracts demonstrated associated cytotoxicity. UPLC-Q/TOF analysis supplemented with the on-line HPLC–DPPH^•^-scavenging method revealed 20 radical scavenging compounds, gallic acid derivatives, most likely, to be major contributors to the overall antioxidant potential of the extracts. Here, cellular antioxidant activity, cytotoxicity, and the inhibition of α-amylase *P*. *officinalis* extracts were reported for the first time. Therefore, *P. officinalis* may be a promising phytochemical to explore for its antidiabetic and antioxidant properties, with future applications in nutraceuticals, cosmeceuticals, and pharmaceuticals. For this purpose, further evaluation should focus on in vivo studies.

## Figures and Tables

**Figure 1 antioxidants-08-00249-f001:**
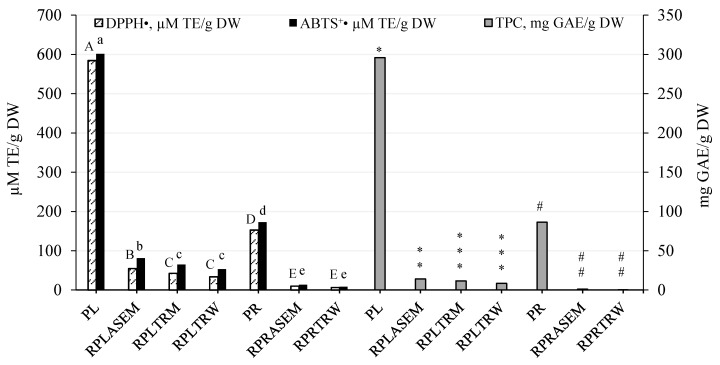
Antioxidant capacity indicators of solid substances determined by the QUENCHER (The QUick, Easy, New, CHEap and Reproducible) method. Values represented as mean ± standard deviation (*n* = 4); the mean values followed by different letters and symbols are significantly different (*p* < 0.05) (ABTS: small letters for leaves (a–c) and for roots (d–e); DPPH^•^: capital letters are used for leaves (A–C) and for roots (D–E) and TPC–symbols are used for leaves (*, **, ***) and for roots (#, ##). TPC is expressed in mg GAE/g DWP; DPPH^•^, ABTS⁺^•^ in μM TE/g DWP. The residues after methanol and water extraction are further referred to by abbreviations composed of mean residue (R), peony (P), leaves or roots (L—leaves, R—roots), and solvent (M—methanol, W—water); ASE and TR mean extraction type: accelerated solvent extraction and traditional extraction, respectively.

**Figure 2 antioxidants-08-00249-f002:**
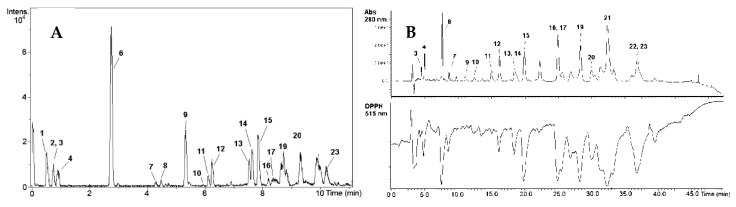
Representative chromatograms of methanol extract (PLASEM) of *P. officinalis*. (**A**) UPLC-Q-TOF chromatogram; (**B**) HPLC-UV-DPPH^•^-scavenging chromatogram showing 19 active compounds (negative peaks at 515 nm), which were detected by comparing their retention times with the UV chromatogram recorded at 280 nm: gallic acid derivatives (3, 4, 5, 6, 7, 10, 12, 14, 15, 16, 17, 19, 20, 21), quercetin derivatives (11, 13), paeoniflorin derivatives (9, 18), and unknown compounds (22, 23).

**Figure 3 antioxidants-08-00249-f003:**
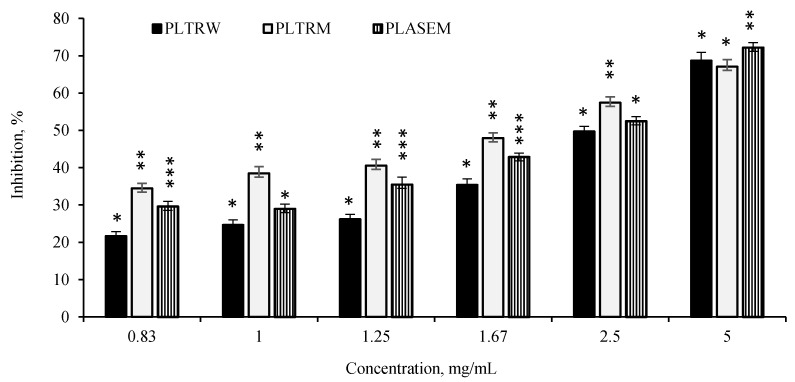
The inhibitory potential of *P. officinalis* methanol and water extracts against porcine α-amylase activity. The values represented as a mean ± standard deviation (*n* = 3). *–***: different symbols indicate significant differences (*p* < 0.05) between different extracts at the same concentration. The abbreviations are composed of the first letter of the plant (P—peony), the botanical part (L—leaves, R—roots), and the solvent (M—methanol, W—water); ASE and TR mean extraction type: accelerated solvent extraction and traditional extraction, respectively.

**Figure 4 antioxidants-08-00249-f004:**
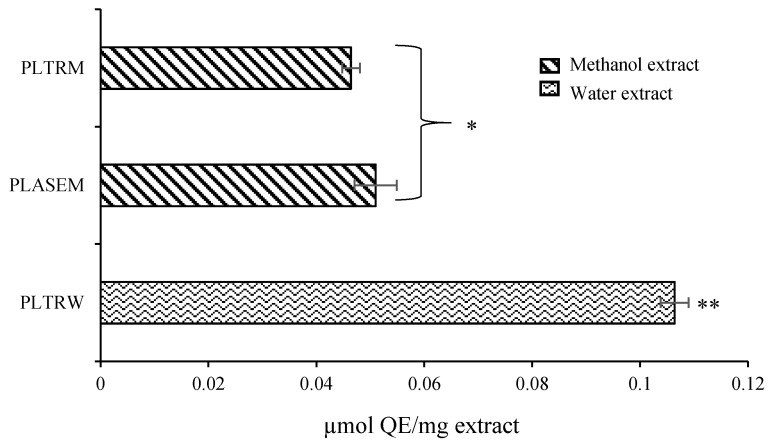
Antioxidant activity of *P. officinalis* extracts evaluated by the cellular antioxidant activity (CAA) method. *–**: the mean ± standard deviation (*n* = 3) values followed by different symbols are significantly different (*p* < 0.05). Other abbreviations are explained in the legend of Figure 3.

**Table 1 antioxidants-08-00249-t001:** Yields, antioxidant capacity, and total phenolic content ^1^ of different *P. officinalis* extracts.

**Samples**	**Yield, %**	**TPC, mg GAE/g**		**DPPH^•^, μM TE/g**		**ABTS^•+^, μM TE/g**	
		**DWE**	**DWP**	**DWE**	**DWP**	**DWE**	**DWP**
*PLASEM*	47.6 ± 1.1 ^a^	516.4 ± 14.53 ^a^	245.6 ± 6.91 ^a^	2424 ± 23.66 ^a^	1153 ± 11.25 ^a^	4524 ± 26.75 ^a^	2151 ± 12.72 ^a^
*PRASEM*	23.9 ± 0.6 ^b^	247.3 ± 8.79 ^b^	59.1 ± 2.1 ^b^	567.7 ± 11.38 ^b^	135.6 ± 2.72 ^b^	935.7 ± 4.51 ^b^	223.5 ± 1.08 ^b^
*PLTRM*	43.5 ± 0.8 ^c^	601.1 ± 10.51 ^c^	285.9 ± 4.99 ^c^	2553 ± 28.40 ^c^	1110 ± 12.34 ^c^	4610 ± 18.70 ^c^	2004 ± 8.13 ^c^
*PLTRW*	33.2 ± 0.2 ^d^	430.6 ± 9.49 ^d^	187.2 ± 4.13 ^d^	2138 ± 13.41 ^d^	710.7 ± 4.46 ^d^	4231 ± 14.07 ^d^	1406 ± 6.76 ^d^
*PRTRW*	19.2 ± 0.5 ^e^	215.7 ± 3.05 ^e^	41.4 ± 0.6 ^e^	343.4 ± 6.06 ^e^	65.94 ± 1.16 ^e^	886.0 ± 7.36 ^e^	170.1 ± 1.41 ^e^
		**ORAC, μM TE/g**		**HOSC, μM TE/g**		**HORAC, µmol CAE/g**	
*PLASEM*		1433 ± 6.93 ^a^	681.4 ± 3.48 ^a^	1957 ± 7.47 ^a^	931.0 ± 9.53 ^a^	1891 ± 13.06 ^a^	899.4 ± 5.45 ^a^
*PLTRM*		1257 ± 10.50 ^b^	546.6 ± 2.59 ^b^	2012 ± 10.22 ^b^	874.6 ± 6.61 ^b^	1758 ± 7.56 ^b^	764.1 ± 8.69 ^b^
*PLTRW*		1232 ± 7.61 ^c^	409.6 ± 1.34 ^c^	2010 ± 8.02 ^b^	668.2 ± 4.15 ^c^	1566 ± 8.45 ^c^	520.3 ± 3.16 ^b^

Values are represented as mean ± standard deviation (*n* = 4); different superscript letters for means down the vertical column that do not share common letters are significantly different (*p* < 0.05). The extracts isolated with methanol and water are further referred to by abbreviations composed of the first letter of the plant (P—peony), the botanical part (L—leaves, R—roots) and the solvent (M—methanol, W—water); ASE and TR mean extraction type: the accelerated solvent extraction and traditional extraction, respectively; DWE—dry weight of extract; DWP—dry weight of plant; ORAC—oxygen radical absorbance capacity; HOSC—hydroxyl radical scavenging capacity; HORAC—(hydroxyl radical antioxidant capacity).

**Table 2 antioxidants-08-00249-t002:** UPLC-Q/TOF identification data of phenolic compounds detected in *P. officinalis* extracts.

Peak No.	Compound	Molecular Formula	t_R_ (min)	*m*/*z*, [M − H]^−^	PLASEM	PRASEM	PLTRM	PLTRW	PRTRW	MS Fragments
1	Quinic acid ^a^	C_7_H_12_O_6_	0.5	191.0563	+	−	+	+	+	-
2	Dihexose ^c^	C_12_H_21_O_11_	0.6	341.1097	+	+	+	+	+	191.0564; 149.0459; 89.0246
3	Galloyl-hexoside ^b,d^	C_13_H_15_O_10_	0.8	331.0671	+	−	+	+	+	169.0140
4	Gallic acid ^a^	C_7_H_5_O_5_	0.9	169.0144	+	+	+	+	+	-
5	Digallic acid ^b,d^	C_14_H_9_O_9_	1.9	321.0253	−	−	−	+	−	169.0138; 125.0240
6	Methyl galate ^b,d^	C_8_H_7_O_5_	2.8	183.0302	+	+	+	+	−	168.0060; 140.0112; 124.0166
7	Tri-galloyl-hexoside ^d^	C_27_H_23_O_18_	4.3	635.0889	+	−	−	−	−	483.0838; 465.0641; 169.0175
8	Unidentified	C_21_H_31_O_13_	4.5	491.1768	+	−	−	−	+	-
9	Paeoniflorin derivative ^b,d^	C_24_H_29_O_13_	5.3	525.1617	+	+	+	+	+	479.1508; 449.1448; 357.1191; 327.1086; 283.0818; 165.0556; 121.0294
10	Tetra-galloyl-hexoside ^b,d^	C_34_H_27_O_22_	5.9	787.1006	+	−	+	+	−	617.0793; 456.0683; 169.0139
11	Quercetin dihexoside ^c,d^	C_27_H_29_O_16_	6.1	609.1460	+	−	+	+	−	463.0883; 301.0325;
12	Quercetin-galloyl-hexoside ^b,d^	C_28_H_23_O_16_	6.2	615.0993	+	−	+	+	−	463.0884; 301.0324; 169.0132
13	Quercetin pentoside ^b,d^	C_20_H_17_O_11_	7.5	433.0779	+	−	+	+	+	301.0340
14	Methyl digallate ^b,d^	C_15_H_11_O_9_	7.6	335.0410	+	+	+	−	+	183.0301; 124.0170
15	Penta-galloyl-hexoside ^b,d^	C_41_H_31_O_26_	7.8	939.1122	+	+	+	+	-	769.0901; 617.0795; 447.0569; 169.0132
16	Isorhamnetin-galloyl-hexosyde ^b,d^	C_29_H_25_O_16_	8.2	629.1149	+	+	+	+	+	477.1046; 315.0568; 169.0141
17	Dihydroxybenzoic acetate-digallate derivative ^b,d^	C_24_H_17_O_15_	8.4	545.0580	+	−	−	−	−	469.0489; 393.0466; 169.0135
18	Paeoniflorin derivative ^b,d^	C_24_H_29_O_13_	8.4	525.1615	−	−	+	+	−	479.1358; 449.1440; 357.1088; 327.1075; 283.0714; 165.0544; 121.0281
19	Dihydroxybenzoic acetate-digallate derivative ^b,d^	C_24_H_17_O_15_	8.7	545.0581	+	+	+	+	−	469.0407; 393.0461; 169.0139
20	Dihydroxybenzoic acetate-digallate derivative ^b,d^	C_24_H_17_O_15_	9.3	545.0582	+	-	+	+	−	469.0381; 393.0468; 169.0140
21	Hexa-galloyl-hexoside ^b,d^	C_48_H_35_O_30_	9.3	1091.1236	+	−	+	+	−	939.1101; 769.0895; 617.0811; 169.0143
22	Unidentified	C_15_H_25_O_26_	9.9	621.0637	+	+	+	+	+	-
23	Unidentified	C_31_H_21_O_19_	10.2	697.0695	+	−	−	+	−	-

^a^ Confirmed by a standard; ^b^ Confirmed by a reference; ^c^ Confirmed by parent ion mass using free chemical database (Chemspider, PubChem); ^d^ Confirmed by MS/MS.

**Table 3 antioxidants-08-00249-t003:** Quinic acid, gallic acid, and quercetin dihexoside recovery (mg/g DWP) and their concentration in the extracts (mg/g DWE) obtained from *P. officinalis* by different solvents.

Sample	Quinic Acid		Gallic Acid		Quercetin Dihexoside *	
	DWP	DWE	DWP	DWE	DWP	DWE
*PLASEM*	3.61 ± 0.12 ^a^	7.58 ± 0.99 ^a^	2.59 ± 0.08 ^a^	5.45 ± 0.09 ^a^	0.50 ± 0.001 ^a^	1.04 ± 0.05 ^a^
*PRASEM*	nd	nd	0.11 ± 0.001 ^b^	0.44 ± 0.001 ^b^	nd	nd
*PLTRM*	1.52 ± 0.09 ^b^	3.50 ± 0.31 ^b^	0.22 ± 0.001 ^b,c^	0.51 ± 0.002 ^b^	0.66 ± 0.003 ^b^	1.52 ± 0.02 ^b^
*PLTRW*	1.83 ± 0.10 ^c^	5.51 ± 0.32 ^c^	4.10 ± 0.25 ^d^	12.33 ± 0.87 ^c^	0.55 ± 0.001 ^c^	1.64 ± 0.01 ^b^
*PRTRW*	0.41 ± 0.001 ^d^	2.14 ± 0.12 ^b^	0.34 ± 0.002 ^b,c^	1.75 ± 0.44 ^d^	nd	nd

nd—not detected; * Based on the calibration curve obtained by using rutin; values represented as mean ± standard deviation (*n* = 3); a–d: means down the vertical column not sharing common letters are significantly different (*p* < 0.05). The extracts isolated with methanol and water are further referred to by the abbreviation composed of the first letter of the plant (P—peony), the botanical part (L—leaves, R—roots), and the solvent (M—methanol, W—water); ASE and TR mean extraction type: accelerated solvent extraction and traditional extraction, respectively.

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
