# Peer review of "Isolation of Strong Antioxidants from Paeonia Officinalis Roots and Leaves and Evaluation of Their Bioactivities"

_antioxidants, 2019, doi:10.3390/antiox8080249_

Round 1

Reviewer 1 Report

The manuscript is dealing with extraction and evaluation of biological active compounds from from Paeonia  officinalis roots and leaves. Manuscript is well written but before publication some modifications are still needed. 

Title is too long and does not reflect the work described in manuscript. In manuscript I couldn  t find informations and studies about valorization as food ingredients.

The purpose of the study should be reformulated.

For Figure 3, I suggest for the axis OX, the concentration of extract to be expressed per gram DW.

Reviewer 2 Report

Data are of interest, but there are some issues and some crucial points that need consideration. The content is sometimes confused and not immediate to understand, mainly because of the several abbreviations used. These should specify at the first mentioned (e.g. DWP, DWE, PLASEM, PRASEM, FL). I suggest major revision of the manuscript before to assess it for publication in Antioxidants. In particular:

Abstract

The abstract should be improved by better explaining the extracts that possess antioxidant activity. On my point of view, it is not sufficient to say, “selected extracts”.

Introduction

In the first part of the introduction section, the authors described the importance of dietary phytochemicals. Is Paeonia officinalis traditional used as food? The authors should clarify this aspect.

Materials and methods

- Page 6, line 234: the concentrations of extracts used for the cytotoxicity assay are too high, ranging from 0.12 to 16.7 mg/ml. Why the authors used these high concentrations?

Results

Results should be explained in a clearer way. Particularly, tables and graphs are not always immediate.

- Table 1, page 7: Why PRASEM and PRTRW extracts were not tested on ORAC, HOSC and HORAC assays?

- Table 1, page 7: the statistical analysis should be better explained in the table legend. This should be due for all the tables and graphs.

- Page 3, line 426: the concentrations of extracts used for the α-amylase inhibition assay are too high. An activity observed at these high concentrations is not relevant.

-Page 4, line 443: Why authors considered only the results obtained after 4 hours of treatment?

Discussion

-          Page 6, lines 514 – 520: It is pretty hard to speak about α-amylase inhibitory activity considering the high concentrations used. Moreover, authors should explain why they assessed this kind of activity and what is the link with antioxidant activity.

-          Page 6, lines 521 – 542: Authors described the potential utility of dietary phytochemicals as anticancer agents and then they concluded that the extracts are not cytotoxic in Caco-2 cells, which are human epithelial colorectal adenocarcinoma cells. This paragraph is contradictory. Please, modify it.

Conclusions

-          Page 7, lines 576 – 577: On the basis of the results, it is not possible to consider the extracts of Paeonia officinalis as cytotoxic and as inhibitors of α-amylase enzyme. Please, revised it.

-          Page 7, lines 578 – 579: From my point of view and considering the results of present manuscript, it is not possible to conclude that Paeonia officinalis may be a promising plant for developing bioactive ingredients with antidiabetic properties.

Round 2

Reviewer 1 Report

The manuscript has been improved and can be published.

Reviewer 2 Report

Dear Editor,

I think that manuscript can be accepted in the present form.